# Numerical Investigation on the Effects of Impeller Structures in Hot Metal Desulfurization Processes by Mechanical Stirring

**Ruizhi Wang [1], Shuyuan Jia [2,3,\*] and Zhu He [2,3,\*]**

1    Baosteel Central Research Institute, Shanghai 201900, China; wangruizhi@baosteel.com
2    The State Key Laboratory of Refractories and Metallurgy, Wuhan University of Science and Technology, Wuhan 430081, China
3    National-Provincial Joint Engineering Research Center of High Temperature Materials and Lining Technology, Wuhan University of Science and Technology, Wuhan 430081, China
\*    Correspondence: jiashuy@wust.edu.cn (S.J.); hezhu@wust.edu.cn (Z.H.)

**Abstract:** With the increasing demand for high-quality steel, the requirements for the efficiency and stability of deep desulfurization are increasing too. The Kanbara Reactor (KR) is widely accepted around the world because of its high efficiency and economy. In order to destroy the rigid motion of molten iron in this area, two kinds of blade structures are designed and compared with traditional blades. In this study, a three-dimensional transient coupling mathematical model was established by using volume of fluid (VOF) and discrete phase model (DPM) to simulate the KR desulfurization process. The turbulence intensity of the molten iron, main vortex size and desulfurizing agent (DA) particle distributions for the three impeller models were investigated in detail. Model results showed that the staggered blade structure may improve the desulfurization efficiency of the KR process, and the desulfurization rate increases from 95.7% to 97.1% when compared with ordinary blades. The main reason for this can be attributed to the enhanced turbulence intensity of the molten iron, larger main vortex size and more uniform DA particle distributions. Plant tests also showed that the desulfurizer consumption per 1 ppm sulfur for the staggered blades was reduced by approximately 8.6%.

**Keywords:** KR desulfurization; impeller structure; numerical simulation; desulfurization efficiency; plant tests

## 1. Introduction

There is a growing demand for high-quality and ultralow sulfur grade steel, particularly for the production of exterior steel plates for high-end vehicles, petroleum pipelines, liquefied natural gas ship plates [1,2], etc. Sulfur in steel will lead to thermal brittleness, and high sulfur content will reduce the corrosion resistance of steel. Therefore, how to realize deep desulfurization of steel has become the key to high-quality steel production [3–6].

Hot metal pretreatment is an important desulfurization step in the steel production process [7]. At present, there are two methods for hot metal pretreatment nodes: the composite injection method and the Kanbara Reactor (KR) method. Due to its good desulfurization kinetic conditions, the latter has the advantages of short treatment time and low operation cost in deep desulfurization, and has become the preferred hot metal deep desulfurization process in the global steel industry [8–10].

In the KR process, a centrifugal force is applied to hot metal in the vessel due to the rotation of the impeller, and the hot metal is forced to move towards the sidewalls. As a result, the surface of the hot metal descends towards the impeller. Low-density particles placed on the hot metal surface also descend to the impeller, collide with it and then disperse in the vessel. To obtain maximum efficiency, the particles must be dispersed in the vessel as widely as possible to increase the total interfacial area, which is primarily controlled by impeller rotation. The dispersion DA particles in the molten

pool depends on the hot metal flow field structure. In order to make the desulfurizer more dispersed and obtain better desulfurization effect, various methods to improve the fluid flow conditions were studied through numerical simulation and experiment. Various methods to improve the fluid flow conditions have been studied by numerical simulations and experiments. For the impeller, the research aspects include the influence of impeller depth [11], impeller speed [12], variable impeller speed [13], eccentric stirring [14] and variable impeller rotation direction [15–17]. These methods can improve the desulfurization efficiency to varying degrees, but at the same time, other issues have occurred, such as increasing power consumption or hot metal free surface sloshing. Some studies believe that the flow field structure of molten iron can be changed to a certain extent by optimizing the impeller structure, so as to improve the dispersion of desulfurizer (DA) particles in molten iron [18–20]. Due to the movement characteristics of molten iron driven by blade rotation of KR stirring, there is a forced vortex zone, which makes the molten iron near the blade shaft maintain rigid rotation movement, and there is no relative movement between liquid micro elements, which constitutes a poor mixing zone.

Therefore, in order to improve the mixing effect in this area, we designed two kinds of blades to destroy the rigid motion in the bad mixing area. Based on the Eulerian–Lagrangian approach of ANSYS Fluent 19.2, the present work presents three different impeller designs and mainly investigates the effects of the impeller structure on the desulfurization efficiency in the KR process. The continuous phase in the ladle is considered to be hot metal and air, and a multiphase flow VOF model was used, which is accomplished by the solution of a continuity equation for the volume fraction of each phase. To achieve coupling between the fluid and impeller, the multiple reference frame model (MRF) method is adopted [21]. The turbulence model used in the calculation was a standard $k$–$\varepsilon$ model, which provided good performance for rotating flows [22]. The DA particles were added via a discrete phase model (DPM), considering the interaction between the flow field and DA particles. At the same time, the desulfurization kinetic model is established by solving the UDS equation to realize the diffusion of sulfr in molten iron and the desulfurization of CaO particles. The desulfurization efficiency of different impellers is quantitatively compared by the established desulfurization kinetic model, which lays a foundation for further optimizing the blade structure and improving the desulfurization efficiency.

## 2. Mathematical Method

### 2.1. Assumptions

In the present work, the following assumptions were made:

(1) The air and hot metal in the ladle were treated as Newtonian fluids with constant physical properties, and the formation of slag in the ladle was ignored [2].
(2) The effect of sulfur content on the interfacial tension was ignored; the constant co-efficient of the hot metal-air and molten metal-DA particles interfacial tension was assumed [14].

### 2.2. Multiple Reference Frame Model

Due to the existence of the rotating impeller, the computational domain was subdivided into: (i) moving and (ii) stationary regions, separated by the interface boundary. The moving reference frame equations described the moving region containing the rotating impeller, while the stationary region was defined by the equations of the stationary reference frame [23–27].

### 2.3. Continuity, Momentum and Energy Equations Used in the Stationary Reference Frame

The continuity, momentum, and energy equations used in the stationary reference frame were as follows [28]:

$$\frac{\partial \alpha}{\partial t} + \nabla \cdot \left( \alpha \vec{v} \right) = 0, \tag{1}$$

$$\frac{\partial(\overline{\rho}\,\vec{v})}{\partial t} + \nabla \cdot (\overline{\rho}\,\vec{v}\,\vec{v}) = -\nabla P + \nabla \cdot \left[\overline{\mu}\left(\nabla\vec{v} + \nabla\vec{v}^{\mathrm{T}}\right)\right] + \overline{\rho}\vec{g} + \vec{F}_{\mathrm{st}} + \vec{S}_{\mathrm{mp}}, \tag{2}$$

$$\frac{\partial(\overline{\rho}\overline{E})}{\partial t} + \nabla \cdot (\overline{\rho}\,\vec{v}\,\overline{E}) = \nabla \cdot (k_{\mathrm{eff}}\nabla T), \tag{3}$$

$$\overline{E} = \overline{h} - \frac{p}{\overline{\rho}} + \frac{\vec{v}^2}{2}, \text{ and } \overline{h} = \int_{T_{\mathrm{ref}}}^{T} \overline{c}_{\mathrm{p}}\mathrm{d}T, \tag{4}$$

where $\alpha$ is the volume fraction of hot metal, $\vec{v}$ is the absolute velocity (m/s), $\overline{\rho}$ is the density of the mixture phase (kg/m$^3$), $\overline{\mu}$ is the viscosity of the mixture phase (Pa·s), $\vec{g}$ is gravity, $\vec{F}_{\mathrm{st}}$ is interfacial tension (N/m$^3$), $\vec{S}_{\mathrm{np}}$ is the source term describing the momentum exchange between the hot metal and DA particles, $\overline{E}$ is the internal energy of the mixture phase (J/m$^3$), $k_{\mathrm{eff}}$ is the effective thermal conductivity (W/(m ·K)), $\overline{h}$ is the sensible enthalpy of mixture phase (J/kg), $T$ is the temperature (K), $T_{\mathrm{ref}}$ is the reference temperature, and $\overline{c}_{\mathrm{p}}$ is the specific heat of the mixture phase at constant pressure (J/(kg ·K)).

*2.4. Continuity, Momentum, and Energy Equations Used in the Moving Reference Frame*

For the moving frame, the respective equations take the following form:

$$\frac{\partial\alpha}{\partial t} + \nabla \cdot \left(\alpha\vec{v}_{\mathrm{r}}\right) = 0, \tag{5}$$

$$\frac{\partial(\overline{\rho}\vec{v})}{\partial t} + \nabla \cdot (\overline{\rho}\vec{v}_{\mathrm{r}}\vec{v}) + \overline{\rho}\left[\vec{\omega} \times \left(\vec{v} - \vec{v}_{\mathrm{t}}\right)\right]$$
$$= -\nabla P + \nabla \cdot \left[\mu\left(\nabla\vec{v} + \nabla\vec{v}^{\mathrm{T}}\right)\right] + \overline{\rho}\vec{g} + \vec{F}_{\mathrm{st}} + \vec{S}_{\mathrm{mp}}, \tag{6}$$

$$\frac{\partial(\overline{\rho}\overline{E})}{\partial t} + \nabla \cdot (\overline{\rho}\vec{v}_{\mathrm{r}}\overline{H} + \overline{\rho}\vec{u}_{\mathrm{r}})$$
$$= \nabla \cdot \left(k_{\mathrm{eff}}\nabla T + \left[\overline{\mu}\left(\nabla\vec{v} + \nabla\vec{v}^{\mathrm{T}}\right)\right] \cdot \vec{v}\right), \tag{7}$$

$$\overline{H} = \overline{h}_{\mathrm{ref}} + \int_{T_{\mathrm{ref}}}^{T} \overline{c}_{\mathrm{p}}\mathrm{d}T, \tag{8}$$

where $\vec{v}_{\mathrm{r}}, \vec{\omega}, \vec{v}_{\mathrm{t}}$ are relative, angular, and translational frame velocities, respectively; $\overline{H}$ is the enthalpy of mixture phase (J/kg), $\overline{h}_{\mathrm{ref}}$ is the sensible reference enthalpy of mixture phase (J/kg), $\vec{u}_{\mathrm{r}}$ is the velocity of the moving reference frame relative to the stationary reference frame, which is derived as follows:

$$\vec{v}_{\mathrm{r}} = \vec{v} - \vec{u}_{\mathrm{r}}, \text{ and } \vec{u}_{\mathrm{r}} = \vec{\omega} \times \vec{r}, \tag{9}$$

*2.5. Turbulence Model*

In the present work, a standard $k$–$\varepsilon$ turbulence model was adopted to treat the turbulence effects, the turbulent kinetic energy equation is [20]:

$$\frac{\partial(\overline{\rho}k)}{\partial t} + \nabla(\overline{\rho}\vec{v}k) = \nabla\left[\left(\overline{\mu} + \frac{\mu_t}{\sigma_k}\right)\nabla k\right] + G - \overline{\rho}\varepsilon, \tag{10}$$

where $\varepsilon$ is the turbulent dissipation rate. The equation is as follows:

$$\frac{\partial(\overline{\rho}\varepsilon)}{\partial t} + \nabla\left(\overline{\rho}\vec{v}\varepsilon\right) = \nabla \cdot \left[\left(\overline{\mu} + \frac{\mu_t}{\sigma_\varepsilon}\right)\nabla\varepsilon\right] + C_1\frac{\varepsilon}{k}G - C_2\overline{\rho}\frac{\varepsilon^2}{k}, \tag{11}$$

$$G = \mu_t\frac{\partial v_i}{\partial x_i}\left(\frac{\partial v_i}{\partial x_j} + \frac{\partial v_j}{\partial x_i}\right), \tag{12}$$

$$\mu_t = \overline{\rho} C_\mu \frac{k^2}{\varepsilon},$$ (13)

where $\mu_t$ is turbulent viscosity, the constant values in the $k$–$\varepsilon$ model are $C_1 = 1.44$, $C_2 = 1.92$ $C_\mu = 0.09$, $\sigma_k = 1.0$, and $\sigma_\varepsilon = 1.3$, respectively.

### 2.6. VOF Method

The volume of fraction (VOF) method was used to track the motion of the interface between air and hot metal. Hot metal and air shared a set of momentum equations described above, and the phase volume fraction $\alpha$, was introduced to track the interface between phases in the cell. The physical properties of the mixed phase, such as density, viscosity, and thermal conductivity, were related to the volume fraction of each phase as follows:

$$\overline{\phi} = \phi_m \alpha + \phi_g (1 - \alpha)$$ (14)

where $\overline{\phi}$, $\phi_m$, $\phi_g$ are the physical properties of the mixture phase, air, and hot metal, respectively. The entire computational domain was used to solve the continuity, momentum, and energy equations and the two phases' volume fraction was updated on real-time scale. The two phases shared the temperature and velocity distributions.

### 2.7. Dispersed Phase Dynamics

Newton's second law is known to govern the motion of a discrete phase DA particle, that is:

$$m_P \frac{dv_d}{dt} = \vec{F}_g + \vec{F}_b + \vec{F}_d + \vec{F}_l + \vec{F}_{vm} + \vec{F}_p$$ (15)

where: $\vec{F}_g$, $\vec{F}_b$, $\vec{F}_d$, $\vec{F}_l$, $\vec{F}_{vm}$ and $\vec{F}_p$ are the gravity, buoyancy, drag, lift, virtual mass, and pressure gradient forces, respectively.

When Newton's second law was transformed into a rotating reference frame, the centrifugal and Coriolis accelerations appear:

$$m_P \frac{dv_d}{dt} = \vec{F}_d + \vec{F}_g + \vec{F}_b + \vec{F}_l + \vec{F}_{vm} + \vec{F}_p + \vec{F}_{centri} + \vec{F}_{Coriolis}$$ (16)

where: $\vec{F}_{centri}$ and $\vec{F}_{Coriolis}$ are the centrifugal and Coriolis forces, respectively. These two forces allowed Newton's laws to be applied to a rotating system since they involved the object motion in an inertial reference frame.

### 2.8. Desulfurization Model

In the pretreatment of hot metal by the KR method, the DA particles reacted with the sulfur element in the hot metal to absorb the sulfur in the hot metal. In addition, the convection and diffusion of sulfur in the hot metal also affected the desulfurization rate of the DA particles. The above phenomenon was taken into account in this study by establishing the desulfurization mass transfer control equations [3,29–31]:

$$\frac{\partial(\overline{\rho}c)}{\partial t} + \nabla \cdot \left( \overline{\rho} \vec{v} c \right) = \nabla \cdot (\overline{\rho} D_S \nabla c) + S_{des}$$ (17)

where $c$ is the concentration of sulfur in the hot metal, and $D_S$ is the diffusion coefficient of sulfur in hot metal, which positively correlates with temperature, we modify the parameters used in [21] to make it more consistent with the actual industrial results:

$$D_S = 6.45 \times 10^{-4} \cdot exp \left( \frac{(3.11T/T_0 - 116) \times 10^{-5}}{R} T \right)$$ (18)

where $T_0$ is the initial molten iron temperature 1573 k, $R$ is the ideal gas constant. The reaction rate constant for a single particle size level can be written as:

$$S_{\text{des}} = \frac{dw[S]}{dt} = -\bar{\rho}\frac{6}{\pi}\frac{\beta_{[S]}}{\delta}\frac{T}{298}c \tag{19}$$

where $S_{\text{des}}$ is the desulfurization rate (kg/s). $d_p$ is the equivalent diameter of the DA particles, The rate of the affected particle size, velocity and local sulfur concentration can be expressed as [31]:

$$Sh = \frac{\beta_{[S]}d_P}{D_S} = 2 + 0.6Re^{\frac{1}{2}}Sc^{\frac{1}{3}} \tag{20}$$

where $Sh$ is the Sherwood number, which represents the ratio of convective mass transfer to diffusion mass transfer rate; $\delta$ is the thickness of CaS. Its growth rate is described in the article of Oeters et al. [30]. Finally, $\beta_{[S]}$ is the sulfur mass transfer rate from the hot metal to CaO particle. $Re$ and $Sc$ are local Reynolds number and Schmidt number, respectively. The DA will form a thicker reaction layer in the hot metal.

During the desulfurization process, the CaS reaction layer will surround the surface of the lime particles. Sulfur slowly diffuses into the lime particles through the growing layer of CaS reaction products. In the case of instantaneous desulfurization reaction with solid lime-based reagent, calcium oxide is converted to sulfide through the following ion exchange reaction [31]:

$$(\text{CaO}) + [S] = (\text{CaS}) + [O] \tag{21}$$

Based on this, the parameters of the desulfurization model were adjusted and optimized by comparing with the results of literature data to make them more consistent with the final desulfurization results. The formation rate and thickness growth rate of the CaS outer layer of the DA were derived as follows:

$$\frac{dw(\text{CaS})}{dt} = \frac{M_{\text{CaS}}}{M_S}\frac{dw[S]}{dt} = \rho_{\text{CaS}}\frac{\pi}{6}\left[d_p{}^3 - (d_p - 2\delta)^3\right] \tag{22}$$

where $M_{\text{CaS}}$ and $M_S$ are the relative molar masses of CaS and S, respectively, and $\rho_{\text{CaS}}$ is the density of CaS.

## 3. Numerical Simulation

### 3.1. Impeller Structure

The effects of three impeller shapes on the desulfurization efficiency in the KR process were investigated as shown Figure 1. All impellers contained four blades that are attached to the shaft at 0°, 90°, 180°, and 270°. Their bottom area was smaller than the top area, resulting in a negative incline from the top to the bottom of the impeller blades. Such an inclination was designed to strengthen the downwards entrainment ability of the impeller. Impeller model 1 is a basic impeller model that is currently used in actual production lines.

Impeller model 2 also has four blades with the same size, but the blades are misaligned in the axial direction. That is, two adjacent blades are arranged one high and one low, as shown in Figure 1b. Impeller model 3 has a pair of large blades and small blades that are alternately and evenly distributed around the mixing shaft. The bottom surfaces of the large blades and small blades are flush with the bottom surface of the mixing shaft. The designs of impeller models 2 and 3 are expected to generate impulse flow that may improve the mixing of DA particles. Table 1 lists the physical properties, geometric parameters, and operating conditions of the fluids used in this study [22].

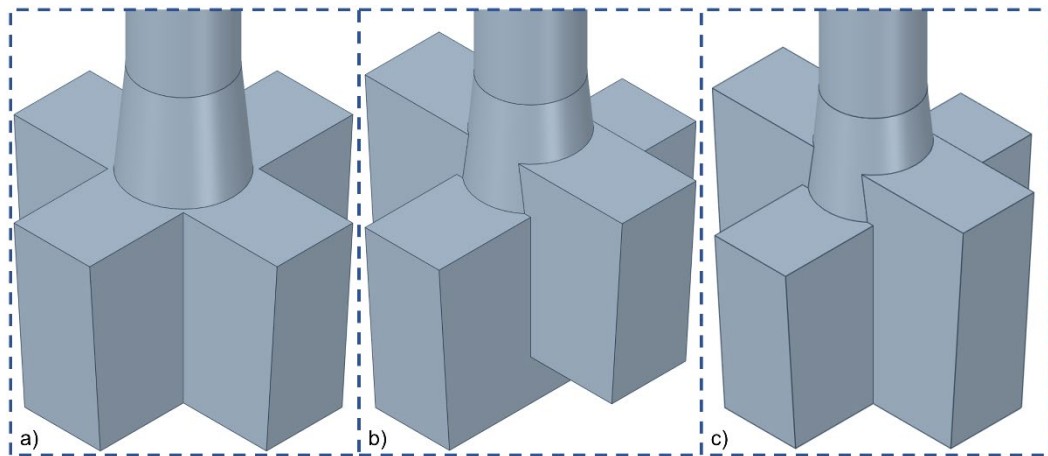

**Figure 1.** Schematic diagram of three blade types. (**a**) Impeller model 1, (**b**) Impeller model 2, (**c**) Impeller model 3.

**Table 1.** Geometric and physical parameters adopted in the numerical simulation.

| Parameter | Value |
|---|---|
| Hot metal density (kg·m$^{-3}$) | 7036 |
| Hot metal viscosity (Pa·s) | 0.0075 |
| Thermal conductivity of hot metal (W/(m·K)) | 36.3 |
| Specific heat of hot metal (J/(kg·K)) | 1.6 |
| Surface tension coefficient of air and molten iron (N/m) | 1.7 |
| Thermal conductivity of DA particles (W/(m·K)) | 3.5 |
| Ladle diameter (mm) | 3856 |
| Ladle height (mm) | 4700 |
| Initial bath depth (mm) | 3621 |
| Impeller height (mm) | 1000/1250(03) |
| Impeller width (mm) | 1470/1600/1632(03) |
| Impeller immersion depth (mm) | 1833 |
| Impeller rotation rate (rpm) | 80 |
| DA particle density (kg·m$^{-3}$) | 3000 |

*3.2. Initial and Boundary Conditions*

The stirring impeller rotates clockwise at a constant speed, and the initial molten iron level L = 3621 mm. For the fluid phase, the top surface of the ladle was set to the pressure-outlet boundary condition, and the atmospheric temperature was assumed to be 373 K. Initially, the mass fraction of sulfur and temperature of the molten iron were set to 0.03% and 1573 K, respectively. The sulfur flux on the top surface, shaft and impeller is considered to be zero. The ladle wall and bottom are treated as fixed walls; the convective heat transfer coefficients applied to the sidewalls and bottom are 55 and 35 W/(m$^2$/K), respectively. DA particles are continuously poured into the container at the mouth of the container, it is assumed that their initial speed and temperature are fixed, and their initial particle size is 0.3 mm. The DA particles are released from the top of the ladle, and the release position is shown in Figure 2b).

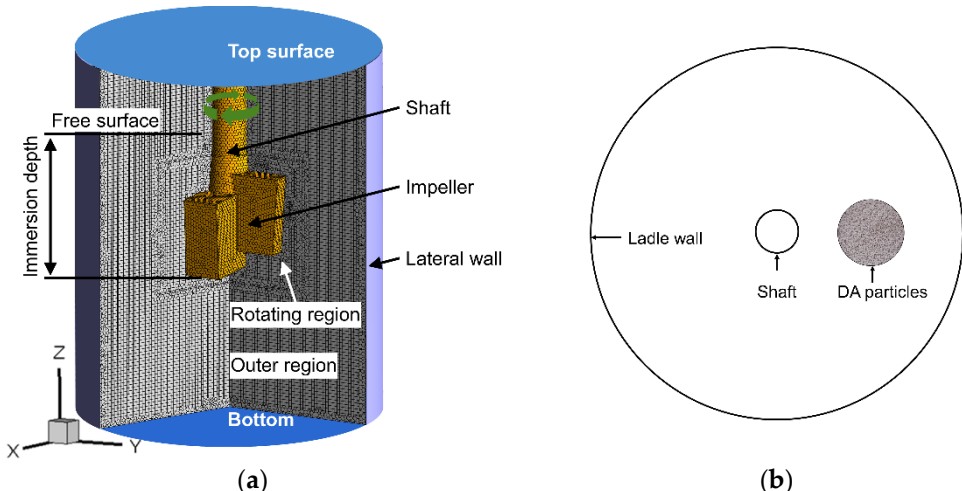

**Figure 2.** Schematic diagram of physical model. (**a**) Schematic diagram of the physical model and mesh; (**b**) DA particle release position.

### 3.3. Numerical Procedure

The CFD simulation studies used the commercial software package Ansys Fluent Version 19.2. The momentum, turbulent kinetic energy, specific dissipation rate, energy, and user-defined scalar equations were discretized using a second-order upwind scheme for higher accuracy. A PISO scheme was utilized for the pressure-velocity coupling. The modified high-resolution interface capturing method was adopted as a discretization scheme for the volume fraction analysis. The convergence criteria for the continuity, momentum, turbulent kinetic energy, specific dissipation rate, user-defined scalar, and volume fraction equations were set at $10^{-3}$, while that for the energy equation was $10^{-6}$. The computational domain is divided into an inner domain and an outer domain, as shown in Figure 2a. The inner domain was set to a rotating reference frame, and the outer domain was set to a stationary reference frame. After the grid sensitivity test, a numerical grid with about 500,000 elements is used. The minimum size of the grid is 20 mm, and the fluctuation of the simulation results relative error is less than 4%. A typical simulation scenario required approximately 800 CPU h with a 0.01 s time step using an AMD Ryzen 9 3900x 12 core processor.

### 3.4. Model Validation

In our previous research [21], the numerical models described above were verified by comparison with the experimental results in the literature [20], which proved the correctness of the present model.

## 4. Results and Discussion

First, the average sulfur mass fraction was calculated, and its evolution with time is shown in Figure 3a. In the first 30–50 s after desulfurizer release, the sulfur content in molten iron decreased slowly. After that, with the continuous addition of desulfurizer and an increasingly uniform distribution of desulfurizer due to stirring, the desulfurization rate increases gradually. From that point, the desulfurization for impeller models 2 and 3 is more efficient than that for impeller 1. After approximately 400 s, the reaction rate decreases due to the decrease in sulfur content and temperature in molten iron. The final desulfurization rates at 600 s are 95.7%, 97% and 97.1% for the three impeller models, as shown in Figure 3b. To determine why impeller models 2 and 3 can improve the desulfurization efficiency, turbulence intensity of the molten iron, main vortex size and DA particle distributions are further studied in the following sections.

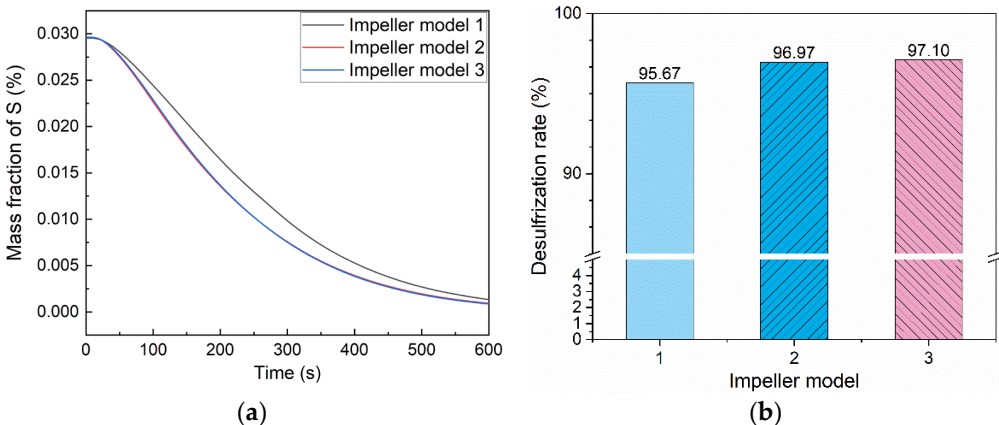

**(a)**

**(b)**

**Figure 3.** Desulfurization efficiency. (**a**) The evolution of average sulfur content with time; (**b**) Desulfurization rate at 600 s.

### 4.1. Characterization of Flow Field

Figure 4 shows the turbulence intensity along the liquid surface and the longitudinal section. The turbulence intensity in the near region of the blades for impeller models 2 and 3 was significantly higher than that for impeller model 1. This is also true for the turbulence intensity on the liquid surface. A higher turbulence intensity on the liquid surface may pull more solid particles down from the surface in small packets into the molten iron [32]. As a result, more DA particles can participate in the desulfurization reaction.

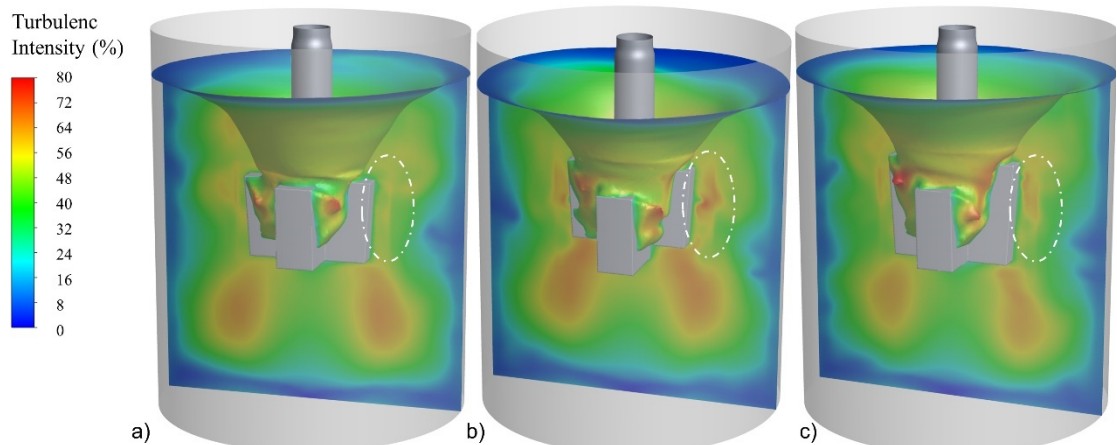

**Figure 4.** The turbulence intensity of the molten iron. (**a**) impeller model 1, (**b**) impeller model 2, (**c**) impeller model 3.

Figure 5 shows the vortex depth (H1) and height (H2) of the main vortex generated by impeller models 1–3. It can be seen that by using impeller models 2 and 3, the strength of the main vortex increases, which is more beneficial for the dispersion of DA particles.

### 4.2. Distribution of DA Particles in Molten Iron

Figure 6 shows the average particle size at t = 600 s. The average particle size for impeller model 3 is 0.392 mm, which is smaller than that for impeller models 1 and 2, indicating that the agglomeration of desulfurizer in the cylindrical rotary zone has been inhibited. The smaller DA particles increase the desulfurization efficiency because of their larger specific surface area.

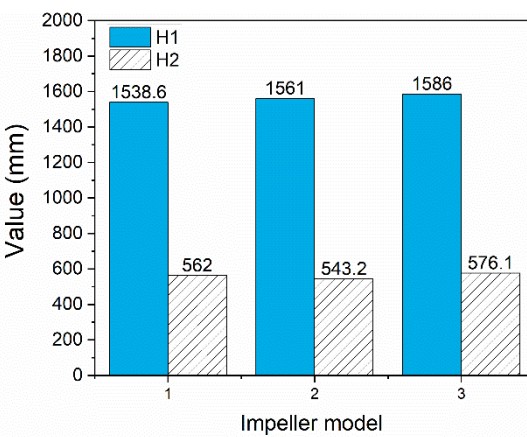

**Figure 5.** The vortex depth and heigh.

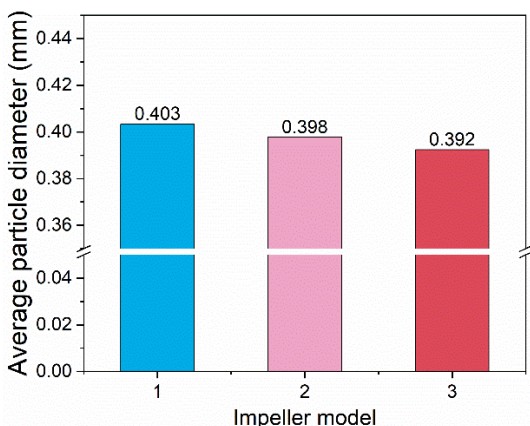

**Figure 6.** Average size of DA particle.

To quantitatively compare the distribution of particles in molten iron, the whole calculation basin is divided into six areas: Internal I1, I2, I3 and external O1, O2, O3, through two planes at different heights, z = 1.7 m and z = 3.2 m, and taking the blade radius as the boundary. The ratio between the amount of desulfurizer and the total amount of desulfurizer in different areas of the molten pool at 600 s is calculated, and the distribution of desulfurizer particles in different areas of the hot metal layer is quantitatively compared, as shown in Figure 7. Because the number of I1 particles is very small, the data are not shown. The number of DA particles in the I2 area for impeller models 2 and 3 is significantly reduced, which makes the probability of DA particle collision decrease in this region. More DA particles entered the O3 region for impeller models 2 and 3, which means that for impeller models 2 and 3, DA particles dispersed more deeply than those for impeller model 1.

### 4.3. Distribution of Desulfurizer

Figure 8 shows the sulfur concentration distributions in the molten iron at 600 s. Obviously, the sulfur concentration distributions are more uniform for impellers 2 and 3. Especially in the dead zone immediately below the impeller, the sulfur concentration is decreased for impeller model 3.

Based on the model results, 100 plant tests were carried out on the impeller made according to the structure of impeller model 3. The results show that compared with impeller model 1, the average stirring time is reduced from 15.7 min to 14.3 min, the average desulfurization rate is 96.13% from 95.05%, and the desulfurizer consumption per ppm sulfur is reduced from 0.347 to 0.317 (about 8.6%), which also proves the effectiveness of the established numerical model.

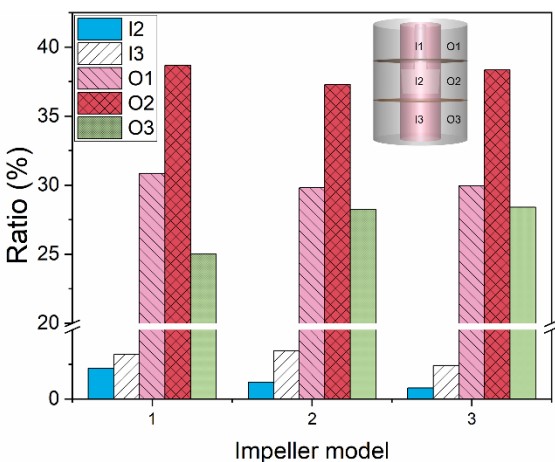

**Figure 7.** Particle distribution for the three impeller models in the ladle.

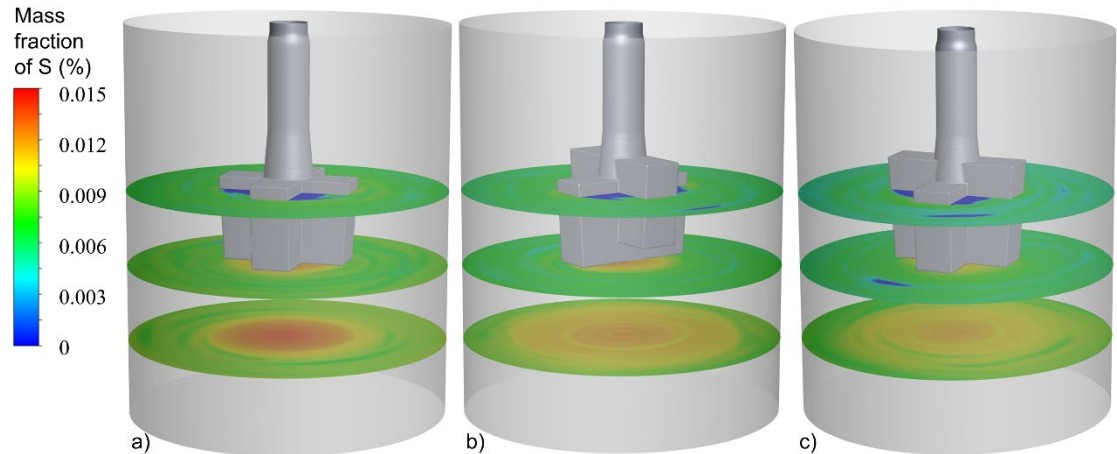

**Figure 8.** Sulfur concentration distribution at 600 s. (**a**) impeller model 1, (**b**) impeller model 2, (**c**) impeller model 3.

## 5. Conclusions

The effects of the impeller structure on molten iron desulfurization with mechanical stirring were investigated by a numerical 3D transient coupled model. The turbulence intensity of the molten iron, main vortex size and DA particle distributions for the three impeller models were investigated in detail. The conclusions are summarized as follows:

1. The staggered blade structure (impeller models 2 and 3) may improve the desulfurization efficiency of the KR process. Compared with ordinary blades (impeller model 1), the desulfurization rate at 600 s increases from 95.7% to 97.1% when impeller model 3 is used.
2. The turbulence intensity of the molten iron, main vortex size and DA particle distributions are enhanced due to the staggered blade structure, which increases the KR desulfurization efficiency.
3. Plant tests proved the validity of the developed numerical models and showed that with impeller model 3, the desulfurizer consumption per 1 ppm sulfur was reduced by approximately 8.6% compared with impeller model 1.

**Author Contributions:** R.W., S.J. and Z.H. conceived and designed the study. S.J. and R.W. accomplished the numerical simulation and data arrangement. The edition work was organized by S.J. under the supervision of Z.H. and all the authors contributed to the discussion about the conclusion. All authors have read and agreed to the published version of the manuscript.

**Funding:** This work was financially supported by the National Natural Science Foundation of China [Grant number 51974211] and the Special Project of Central Government for Local Science and Technology Development of Hubei Province [Grant numbers 2019ZYYD003, 2019ZYYD076].

**Data Availability Statement:** Data is contained within the article.

**Conflicts of Interest:** The authors declare no conflict of interest.

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
