# Peer review of "Numerical Investigation on the Effects of Impeller Structures in Hot Metal Desulfurization Processes by Mechanical Stirring"

_metals, doi:10.3390/met12020229_

Round 1

Reviewer 1 Report

REVIEW

on article

Numerical investigation on the effects of impeller structures in hot metal desulfurization process by mechanical stirring

Ruizhi WANG, Shuyuan JIA and Zhu HE

SUMMARY

At present, in the steelmaking industry, reducing the sulfur content in pig iron scrap is a serious scientific problem. Sulfur removal is a rather complex metallurgical task, the solution of which is associated with significant expenditures of energy carriers and reagents, which can lead to an increase in the cost of products, and often make products unprofitable. According to various sources, the cost of removing sulfur reaches 2%. All this sets the task of developing and introducing new resource-saving technologies. One of such technologies is metal desulfurization in induction furnaces. This becomes relevant in connection with the appearance in the industry of induction melting plants with wide and technological possibilities for obtaining alloys of ferrous and non-ferrous metals.

The authors of the article, based on the Navier-Stokes equations and the continuity equations, applied classical turbulence models and developed an algorithm for the numerical simulation of desulfurization processes. The results of numerical analysis in the form of fluid velocity fields are obtained and graphically executed.

The list of references consists of 25 sources.

Overall, the article made a good impression. However, there are inaccuracies that require clarification.

COMMENTS.

  1. The authors must redo the Abstract and bring it in compliance with the requirements of the Metals journal. The scientific problem is not described (Background). The Abstract should comprise up to 200 words. The scientific novelty is not indicated. Editors strongly encourage authors to use the following style of structured abstracts, but without headings: (1) Background: Place the question addressed in a broad context and highlight the purpose of the study; (2) Methods: Describe briefly the main methods or treatments applied; (3) Results: Summarize the article's main findings; and (4) Conclusions: Indicate the main conclusions or interpretations. The abstract should be an objective representation of the article.
  2. The main aim of the article is indicated unclear. The article largely repeats the previous article by the authors [11] Li, Q.; Ma, S.W.; Shen, X.Y.; Li, M.M.; Zou, Z.S. Effects of Impeller Rotational Speed and Immersion Depth on Flow Pattern, Mixing and Interface Characteristics for Kanbara Reactors using VOF-SMM Simulations. Metals, 2021, 11, 1596.
  3. The Introduction section is poor. The authors must expand the Introduction section following the current actual literature review and detailed analysis of the article's contribution.
  4. Authors must bring references in accordance with the requirements of the Metals journal. In the text, reference numbers should be placed in square brackets [ ] and placed before the punctuation.
  5. Based on the analysis of the literature, the authors in the final paragraph of the Introduction section should formulate the purpose of the study and highlight the novelty of the article.
  6. Write down the boundary conditions of the problem mathematically. They are unclear in the text of the article.
  7. The numerical simulation procedure and the algorithms used are poorly described. I recommend highlighting this in more detail.
  8. I recommend that the authors in the Discussion section compare the results obtained by authors with other researchers' data and give a deep analysis of the results obtained.
  9. I recommend expanding the article with a more detailed and in-depth analysis of the results.

In general, the article is devoted to an interesting and topical problem of modeling desulfurization processes in steelmaking production. I recommend the article for publication after appropriate corrections.

Author Response

Point 1: The authors must redo the Abstract and bring it in compliance with the requirements of the Metals journal. The scientific problem is not described (Background). The Abstract should comprise up to 200 words. The scientific novelty is not indicated. Editors strongly encourage authors to use the following style of structured abstracts, but without headings: (1) Background: Place the question addressed in a broad context and highlight the purpose of the study; (2) Methods: Describe briefly the main methods or treatments applied; (3) Results: Summarize the article's main findings; and (4) Conclusions: Indicate the main conclusions or interpretations. The abstract should be an objective representation of the article.

Response 1: Thank you. According to your suggestion, the summary has been modified, and the modified part is marked in red font.

Point 2:The main aim of the article is indicated unclear. The article largely repeats the previous article by the authors [11] Li, Q.; Ma, S.W.; Shen, X.Y.; Li, M.M.; Zou, Z.S. Effects of Impeller Rotational Speed and Immersion Depth on Flow Pattern, Mixing and Interface Characteristics for Kanbara Reactors using VOF-SMM Simulations. Metals, 2021, 11, 1596.

Response 2: Thanks for your question. The main aim of our article is different from the previous article by the authors Li, Q. Li, Q mainly explores the influence of different rotating speeds and immersion depth on the flow field, and our purpose is based on the following reasons: Since Kr method desulfurizes molten iron by stirring, there is bound to be a forced vortex zone near the shaft of the blade, and there is a lack of relative movement between the fluids at this position, which is unfavorable to the dispersion of particles, Therefore, we designed two kinds of blades to reduce this problem and improve the dispersion of desulfurizer. At the same time, we designed a desulfurization kinetic model to quantitatively calculate the desulfurization effect.

Point 3:The Introduction section is poor. The authors must expand the Introduction section following the current actual literature review and detailed analysis of the article's contribution.

Response 3: Thank you for your suggestion. I expanded the introduction to make it more logical and highlight the focus of our work.

Point 4:Authors must bring references in accordance with the requirements of the Metals journal. In the text, reference numbers should be placed in square brackets [ ] and placed before the punctuation.

Response 4: Thank you. The references have been revised as required.

Point 5: Based on the analysis of the literature, the authors in the final paragraph of the Introduction section should formulate the purpose of the study and highlight the novelty of the article.

Response 5: Thank you for your suggestion. I revised the introduction and emphasized the novelty of our work.

Point 6:Write down the boundary conditions of the problem mathematically. They are unclear in the text of the article.

Response 6: Thank you. In order to save the length of the article and avoid repetition, we have reduced the description of boundary conditions and given them in the form of references, but it may lead to inconvenient reading. Therefore, some boundary conditions are added.

Point 7:The numerical simulation procedure and the algorithms used are poorly described. I recommend highlighting this in more detail.

Response 7: Thank you for your suggestion. I added references corresponding to mathematical models.

Point 8:I recommend that the authors in the Discussion section compare the results obtained by authors with other researchers' data and give a deep analysis of the results obtained.

Response 8: Thank you for your suggestion, but there are some differences in the results based on the size difference of the model. We have not made this comparison. however we have conducted some industrial tests for verification. For confidentiality reasons, the detailed data of industrial tests cannot be provided. At the same time, some results of industrial test results are added at the end of the article

Point 9:I recommend expanding the article with a more detailed and in-depth analysis of the results.

Response 9: Thank you for your suggestion. We added the industrial test results as a supplement at the end of the article.

Reviewer 2 Report

Dear Authors,

I cannot recommend your article for publication, because the paper does not form a coherent work.

The mathematical model is described too laconically. The mathematical model lacks references to literature. There is no definition of many variables: average viscosity, density, etc.

There is no information on what studies were used to determine equations (14), (15), (16), and (18). There is no information about many of the physico-chemical parameters adopted in the numerical analyses. Lack of parameters for the turbulence model. No information on the types of finite elements used and their sizes.

In my opinion, due to the distinctly different physico-chemical properties of water and molten metal, validation of the mathematical and physical model of the phenomenon based on water experiments [20] is not appropriate.

Moreover:

  • different font sizes,
  • article not correctly formatted
  • Fig. 8 lacks description of subfigures.

Author Response

Response to Reviewer 2 Comments

Point 1: The mathematical model is described too laconically. The mathematical model lacks references to literature. There is no definition of many variables: average viscosity, density, etc.

Response 1: Thank you for your suggestion. The mathematical model we use is not complex. We have added references to the mathematical model, especially the desulfurization kinetic model. The two-phase flow includes molten iron and air. Since the density of molten iron is much higher than that of air, the change of temperature and physical parameters of air are ignored in this study. In addition, some physical quantities are added in Table 1 for easy reading.

Point 2: There is no information on what studies were used to determine equations (14), (15), (16), and (18). There is no information about many of the physico-chemical parameters adopted in the numerical analyses. Lack of parameters for the turbulence model. No information on the types of finite elements used and their sizes.

Response 2: Thank you for your suggestion. References 14-16 have been added to the equation.

We use the finite difference method and solve it based on the pressure solver. At the same time, the minimum mesh size is added in 3.3 numerical procedure.

Point 3: In my opinion, due to the distinctly different physico-chemical properties of water and molten metal, validation of the mathematical and physical model of the phenomenon based on water experiments [20] is not appropriate.

Response 3:Thank you for your comments. Due to the problems that the molten iron ladle is invisible and it is difficult to study the characteristics of flow field in high temperature test, it is a common research method to build water model test through geometric similarity and dynamic similarity in this field. It is true that the 1:1 model is closer to the actual situation, but it has not been respected because of its floor area and economy.

Point 4: different font sizes, article not correctly formatted Fig. 8 lacks description of subfigures.

Response 4: Sorry to have these questions, and thank you for your suggestions. I have revised these problems in the text.

Reviewer 3 Report

The authors presenta a paper of interest for iron refining. The paper is, overall acceptable and well explained though, some aspects must be carefully considered and, evntually, correscted.

  1. On page 4, lines 139-140, a weird English expression is used to expres that beta is the mas transfer coefficient. I do not know where from this expression comes from but please say that it is mass transfer coefficient.
  2. I can see that you use the Lagrange model including the main forces acting on a particle. However, you did not report  any group of trajectories of partciles that, according to your comments, must be influenced by the impeller design, You please elaborate on this.
  3.  On lines 255-56, page 8, you mention that the impeller 3 provides an improved agglomeration of particles. Well, I do not see any populations model around here, Thus, from where do you reach this observation?
  4. Overall the kinetic differences among the three impellers are small and can easily be masked by the slag, temperature gradients, etc. in the real process. My point is that from a practical standpoint, there is not diffreneces using any of these three impellers. I suggest you to make a serious discussion on this regard.

Author Response

Response to Reviewer 3 Comments

Point 1: On page 4, lines 139-140, a weird English expression is used to expres that beta is the mas transfer coefficient. I do not know where from this expression comes from but please say that it is mass transfer coefficient.

Response 1: Thank you for your suggestion. I have added references to the equations of desulfurization model.

Point 2: I can see that you use the Lagrange model including the main forces acting on a particle. However, you did not report  any group of trajectories of partciles that, according to your comments, must be influenced by the impeller design, You please elaborate on this.

Response 2: Thank you for your suggestion. We have made streamline results, but because this result is used in the analysis of flow field. Due to the height drop of the blade structure, part of the molten iron will move towards the wall of the molten iron ladle. When rotating to the lower blade, it tends to cross the top surface of the blade, which produces axial flow.

Point 3: On lines 255-56, page 8, you mention that the impeller 3 provides an improved agglomeration of particles. Well, I do not see any populations model around here, Thus, from where do you reach this observation?

Response 3: Thanks, we have made statistics on the particles in different areas. The results show that the proportion of O2 particles in the area near the blade is reduced, and the average diameter of particles is also reduced, which shows that the agglomeration is improved.

Point 4: Overall the kinetic differences among the three impellers are small and can easily be masked by the slag, temperature gradients, etc. in the real process. My point is that from a practical standpoint, there is not diffreneces using any of these three impellers. I suggest you to make a serious discussion on this regard.

Response 4: Thank you for your suggestion. We have made an industrial test on No. 3 impeller, and the test results show that it is caused by.

Round 2

Reviewer 1 Report

All my comments were taken into account and the necessary corrections were made in the text of the article. The article looks much better.
One small note.

References to literary sources in square brackets should be written not as an upper script, but as a normal text, for example, [1, 2-5] and so on.

Author Response

All my comments were taken into account and the necessary corrections were made in the text of the article. The article looks much better.

One small note.

References to literary sources in square brackets should be written not as an upper script, but as a normal text, for example, [1, 2-5] and so on.

Response: Thank you for your suggestions, which have greatly helped to improve this article. According to your request, I modified the format of references.

Reviewer 2 Report

Dear Authors,

By fluctuation of parameters of numerical analyses I mean a change of the most important parameters of a given analyzed system. In the case of the KR reactor, this will be the concentration of S. Changes in results at the level of 4% (S concentration) make it probably impossible to determine which of the analyzed models is the best. Please determine the measurement uncertainties of the most important parameters.

The mathematical model is still described in the wrong way.

What equations are described by Smp, Fst, drag, lift, and virtual mass forces? The drag force in most cases depends on the velocity.

Lack of parameters for the turbulence model.

Why were these initial values of sulfur in the melt and DA particle size adopted?

If equation (18) is taken from the paper [32], the citation to this literature should be repeated.

The authors refer to a publication [22] that I could not find.

Discussion concerning Point 3.

Everyone is aware of the problems associated with validating or verifying such a numerical model. The effectiveness of a given KR reactor could be indirectly determined by, for example, analyzing the composition of the steel after solidification

Moreover:

  • there should be a comma or a period at the end of the equations
  • l. 12 is “Kanbara reactor” should be “Kanbara Reactor”
  • l. 37 is “kanbara reactor (KR)” should be “Kanbara Reactor”
  • l. 102 and other - without indentation
  • l. 102 and other - a bad font, too much spacing
  • equation (7) the variable ur should be denoted as a vector
  • l. 159 is “… is the thickness of CaS, ...” The equation (17) together with the description should be before this line
  • l. 262 and other - is “depth(H1) and height(H2) …” should be "depth (H1) and height (H2) …”

Author Response

Response to Reviewer 2 Comments

Response:Thank you for your detailed suggestions and questions.

Point 1:

By fluctuation of parameters of numerical analyses I mean a change of the most important parameters of a given analyzed system. In the case of the KR reactor, this will be the concentration of S. Changes in results at the level of 4% (S concentration) make it probably impossible to determine which of the analyzed models is the best. Please determine the measurement uncertainties of the most important parameters.

Response:The 4% here refers to the relative percentage difference of results obtained under different grid parameters. It does not mean that the absolute value of s content is 4%.

Point 2:

The mathematical model is still described in the wrong way.

What equations are described by Smp, Fst, drag, lift, and virtual mass forces? The drag force in most cases depends on the velocity.

Lack of parameters for the turbulence model.

Response 2:Thank you for your suggestions. The turbulence model and particle force are not rewritten with UDF, so many parameters are the default values, and the description at the equation is simplified.

These force equations of particle are basic equations and have no parameters to be defined. At the same time, they are mentioned in some references, such as reference [14], so we do not describe them in this paper.

We added turbulence model equations and parameters.

Point 3:

Why were these initial values of sulfur in the melt and DA particle size adopted?

Response 3:The initial sulfur content and particle diameter are derived from factory production data. The sulfur content takes an average sulfur content before desulfurization, and the particles are the same.

Point 4:

If equation (18) is taken from the paper [32], the citation to this literature should be repeated.

Response 4:No, equation (18) does not come from the literature, but is derived from the law of conservation of mass. The equation expresses that the consumption of S and the formation of CAS in molten iron are related to their molar mass.

Point 5:

The authors refer to a publication [22] that I could not find.

Response 5:It is a Chinese article. Here is the website of the literature.

http://dx.doi.org/10.13228/j.boyuan.issn0449-749x.20170554

Point 6:

Discussion concerning Point 3.

Everyone is aware of the problems associated with validating or verifying such a numerical model. The effectiveness of a given KR reactor could be indirectly determined by, for example, analyzing the composition of the steel after solidification.

Response 5:

Yes, you are right in your view of verifying the mathematical model from the water model. Indeed, there are differences in physical and chemical properties between molten iron and water.

For the flow structure, it is difficult to observe the flow pattern of the thermal model in the existing technology, so we can only use the similarity principle to verify the accuracy of the mathematical model by verifying the vortex structure of the water model test.

For the desulfurization model, many researchers use a large amount of data to predict hot metal desulfurization through data-driven method, or through dynamic model. There are more or less working condition differences and other problems, which may lead to problems in applicability.

The desulfurization model used in our article is in good agreement with the results of the plant. However, the focus of this paper is not to compare the desulfurization model, but to improve the desulfurization efficiency brought by this new blade structure. At the same time, we have also conducted hundreds of factory practical tests. From the average desulfurization effect, it is confirmed that the desulfurization effect of No. 3 propeller is better.

As for the progress of desulfurization model, we think it needs to be further optimized and compared with the results of the system.

Point 7:

Moreover:

there should be a comma or a period at the end of the equations

  1. 12 is “Kanbara reactor” should be “Kanbara Reactor”
  2. 37 is “kanbara reactor (KR)” should be “Kanbara Reactor”
  3. 102 and other - without indentation
  4. 102 and other - a bad font, too much spacing

equation (7) the variable ur should be denoted as a vector

  1. 159 is “… is the thickness of CaS, ...” The equation (17) together with the description should be before this line
  2. 262 and other - is “depth(H1) and height(H2) …” should be "depth (H1) and height (H2) …”

Response 7:

Thank you for your careful reading and I have revised these questions.

Round 3

Reviewer 2 Report

Dear Authors,

a few more comments and questions:

  • no literature reference for the turbulence model and the constants used for the model
  • l. 318 is “Heat is lost to surrounding from top surface in the form of natural convection and radiation …” - no radiation model in the article

Discussion concerning Point 4

Obviously, that equation (18) is derived from the law of conservation of mass, but the constants appearing in this equation were obtained from experimental studies.

Therefore, please describe this research or cite the literature you used.

Author Response

Dear reviewer:

Thank you for your question. I have marked the revised position in red.

Point 1

no literature reference for the turbulence model and the constants used for the model

  1. 318 is “Heat is lost to surrounding from top surface in the form of natural convection and radiation …” - no radiation model in the article

Response: Thank you very much for asking this question. The description of boundary conditions "heat is lost to surrounding from top surface in the form of natural construction and radiation..." is inappropriate. In order to simplify the calculation, we ignore the radiant heat transfer and enhance the convective heat transfer to make up for the neglected radiant heat.

The temperature drop of the plant is about 45 K, which is consistent with the temperature drop results obtained by our calculation.

 Point 4

Obviously, that equation (18) is derived from the law of conservation of mass, but the constants appearing in this equation were obtained from experimental studies.

Therefore, please describe this research or cite the literature you used.

Response: Thank you for explaining again. I thought you mentioned equation (22), whatever. I modified the description of the position of this equation and added references.
